# Risky behavior during stair descent for young adults: Differences in men versus women

HyeYoung Cho[¤a☯], Amanda J. Arnold[¤b‡], Chuyi Cui[¤c‡], Zihan Yang[¤d‡], Tim Becker[¤e‡], Ashwini Kulkarni[‡], Anvesh Naik[‡], Shirley Rietdyk[*☯]

Department of Health and Kinesiology, Purdue University, West Lafayette, IN, United States of America

☯ These authors contributed equally to this work.
¤a Current address: Department of Kinesiology, University of Northern Iowa, Cedar Falls, IA, United States of America
¤b Current address: Department of Health, Exercise, and Sport Studies, Denison University, Granville, OH, United States of America
¤c Current address: Department of Neurology and Neurological Sciences, Stanford University School of Medicine, Stanford, CA, United States of America
¤d Current address: Fashion Accessory Art and Engineering College, Beijing Institute of Fashion Technology, Beijing, CHINA
¤e Current address: School of Medicine, Indiana University, Indianapolis, IN, United States of America
‡ AJA, CC, ZY, TB, AK and AN also contributed equally to this work.
* srietdyk@purdue.edu

**Data Availability Statement:** The dataset is publicly available at Purdue University Research Repository. https://purr.purdue.edu/publications/4218/1.

## Abstract

Injuries commonly occur on stairs, with high injury rates in young adults, especially young women. High injury rates could result from physiological and/or behavioral differences; this study focuses on behaviors. The purposes of this observational study were (1) to quantify young adult behaviors during stair descent and (2) to identify differences in stair descent behavior for young adult men versus women. Young adult pedestrians (N = 2,400, 1,470 men and 930 women) were videotaped during descent of two indoor campus staircases, a short staircase (2 steps) and a long staircase (17 steps). Behaviors during stair descent were coded by experimenters. Risky behaviors observed on the short staircase included: No one used the handrail, 16.1% used an electronic device, and 16.4% had in-person conversations. On the long staircase: 64.8% of pedestrians did not use the handrail, 11.9% used an electronic device, and 14.5% had in-person conversations. Risky behaviors observed more in women included: less likely to use the handrail (long staircase), more likely to carry an item in their hands (both staircases), more likely to engage in conversation (both staircases), and more likely to wear sandals or heels (both staircases) ($p \leq 0.05$). Protective behaviors observed more in women included: less likely to skip steps (both staircases), and more likely to look at treads during transition steps (long staircase) ($p \leq 0.05$). The number of co-occurring risky behaviors was higher in women: 1.9 vs 2.3, for men vs women, respectively ($p < 0.001$). Five pedestrians lost balance but did not fall; four of these pedestrians lost balance on the top step and all five had their gaze diverted from the steps at the time balance was lost. The observed behaviors may be related to the high injury rate of stair-related falls in young adults, and young women specifically.

**Funding:** The authors received no specific funding for this work.

**Competing interests:** The authors have declared that no competing interests exist.

## Introduction

Falls often occur due to environmental hazards [1] and stairs are one of the most serious hazards encountered regularly in daily activities [2, 3]. Although less than 1% of the waking day is spent on stairs, 12% of falls in young adults occurred while navigating stairs [4, 5]. Falls on stairs are more likely to result in injuries [6–12], with 10% of fatal falls being associated with stair use [13].

Falls on stairs occur across the lifespan [14–16]; average injury rate of stair-related falls in the United States demonstrates a trimodal distribution with peaks at $\leq 3$ years of age, young adults in their 20's, and adults $\geq 85$ years [16]. Higher injuries could result from physiological factors (e.g., decreased strength in older adults [17]), and/or behavioral factors (e.g., not using the handrail [18]). Since physiological parameters are presumably not compromised in young adults, it seems that quantifying behavior will be highly relevant to understand fall risk on stairs.

While both men and women demonstrated a peak in injury rate in their 20's, there are two observations that emphasize that young women are particularly susceptible to injury on stairs. First, the injury rate for young women is approximately 80% higher than the rate for young men [16]. Second, the injury rate is highest for women in their 20's relative to all other age decades (for both sexes) with the exception of women $\geq 81$ years [16]. However, the reason women in their 20's sustain more stair-related injuries is unknown. The focus of this study is to quantify behaviors observed in young adults that may contribute to fall risk on stairs, and to determine if the higher injury rate in young women versus men is related, at least partially, to behavioral differences.

Fall risk on stairs can be increased by a range of behaviors such as: not using the handrails, using the hands for other tasks such that the handrail cannot be grasped if a person stumbles, not visually attending the stairs, using electronic devices and other distracting activities, skipping steps, and wearing inappropriate footwear. Observational research has demonstrated that the majority of pedestrians do *not* use the handrails (57–94%) [18–22]. Gaze behavior is critical for safe stair traversal [23]. Video coding of a small group of pedestrians (from security video footage) indicated that not looking down at the step appeared to be associated with falls [24]. Furthermore, multitasking affects situational awareness and gait behavior [25], which may result in falls [5] and injuries [26]. Lab-based research has demonstrated that talking on a mobile phone impairs stair gait [27], and phone use strongly draws visual attention away from the stairs [28, 29]. Observational research indicates that those who use mobile devices on stairs were less likely to use the handrail and were more likely to drift across the stairway [22]. Identifying sex-related differences in these behaviors may reveal why women sustain more stair related injuries [16].

The purposes of this observational study were (1) to identify stair behaviors in young adults that may increase fall risk and (2) to determine if young women are more likely to demonstrate these risky behaviors. Since the majority of falls occur during descent [7, 9, 12, 13], we video-taped and coded stair descent of pedestrians. Indoor staircases were selected since the majority of stair-related falls in young adults occurred indoors (69%) [5]; staircases in university campus buildings were selected to ensure the majority of observed pedestrians were young adults. We videotaped both a short staircase (2 steps) and a long staircase (17 steps), as there is a high accident rate on staircases with five or fewer steps, but falls from higher heights result in more injuries [9, 30].

## Methods

The study was approved by the University Institutional Review Board (IRB). The IRB determined that the study met the criteria for exemption since observations occurred in a public

setting and we did not obtain any personal information from the pedestrians. No informed consent was required.

## Staircases

The short staircase was located indoors and had two steps (Fig 1A and 1B). The lower level and step were carpeted with a low-pile gray carpet (multiple small loops that are not cut, sometimes called Berber carpet). The upper level was carpeted with a geometric patterned yellow and gray Berber carpet. The risers were black. The tread width was 173.0 cm. Both treads were 41.6 cm in depth and both risers were 10.2 cm in height. The treads had a square metal stair nosing that was 2 x 5 cm (height by depth); the nosing was almost flush with the tread and riser. Five black friction strips (0.7 cm wide) were embedded in the nosing. The first strip was placed 0.2 cm from the tread edge. No stair highlighters were present, although the metal nosing may act as a highlighter. The handrail was round with diameter 4.8 cm and located 93.2 cm above the stairs. No windows were located near the short staircase, so the lighting level was only affected minimally by outdoor light levels; the light level was 125 lux (daytime) and 112 lux (evening). Evening light levels were recorded because videotaping occurred in the evening.

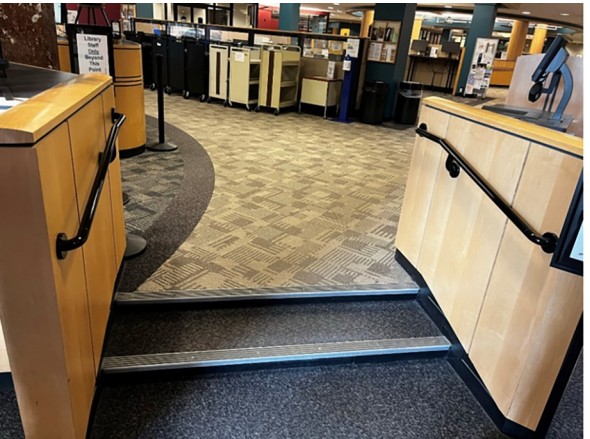
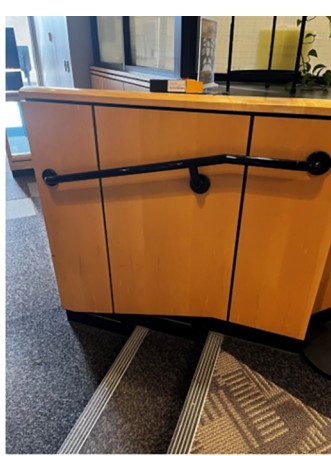

A                                                    B

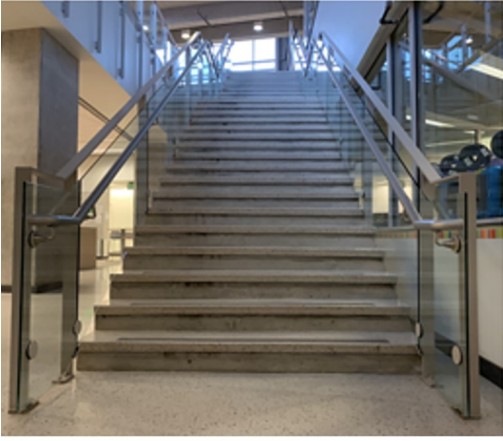
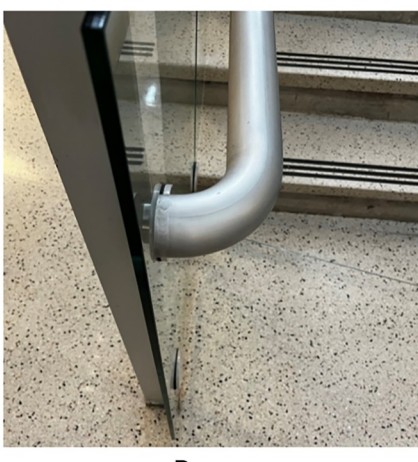

C                                                    D

**Fig 1. Photos of the two staircases.** The short staircase (two steps) (A, B) and long staircase (17 steps) (C, D). The long staircase (34 steps) included a landing, the video was coded from the landing to the bottom of the stairs (17 steps).

The long staircase was located indoors and had 34 steps (Fig 1C and 1D). There was a landing (121.9 cm long) separating the top 17 steps from the bottom 17 steps. Because pedestrians were not sufficiently visible for coding on the top set of stairs, only the bottom 17 steps were coded. The flooring was the same on the landing, treads, and lower level: smooth finish, beige color with brown and black flecks. The risers were smoothed cement. The tread width was 182.0 cm. Tread depth ranged from 29.2 to 30.5 cm. Deeper treads were observed on the top to middle steps (steps 1–15 were 30.5 cm). Shallower treads were observed at the bottom steps (step 16 was 30.2 cm and step 17 was 29.2 cm). The risers ranged in height from 15.7 to 16.5 cm. The shortest risers were found at the top of the staircase (step 2 was 15.8 cm and step 3 was 15.7 cm), and taller risers occurred at the bottom (step 15 was 16.3 cm, step 16 was 16.2 cm, and step 17 was 16.5 cm). The remainder of the risers (step 1 and steps 4–14) were 16.0 cm. The stair nosing was made of the same material as the treads and was square in shape with a slope where the underside of the nosing joined the riser, termed bevel nosing. Nosing height was 3.2 cm, and depth was 1.2 cm (measured horizontally from front of nosing to riser). Three black friction strips (0.5 cm wide) were placed on each tread; the first strip was placed 1.3 cm from the tread edge. No stair highlighters were present, although the friction strips may act as a highlighter that was not flush with the front edge of the stair. Windows were located near the top of the long staircase (Fig 1), influencing the lighting level. Light levels were measured for two extremes: a sunny day and an overcast day (no evening videos were recorded on the long staircase). During the sunny day, the lighting level at the top of the stair was 278 lux and decreased monotonically to 120 lux at step 17. During the cloudy day, the lighting level at the top step was 122 lux and decreased monotonically to 68 lux at step 17.

## Data collection

To record the videos, smartphone cameras were used (iPhone and Nokia). The cameras were placed unobtrusively at the bottom and to the side of the staircases. The camera position captured a frontal view of the pedestrian, such that the face and full body were visible for video coding. Due to the extensive use of electronic devices with videos, video recording in public settings is generally acceptable [21]. An experimenter was seated by the camera, was engaged in an activity (e.g., working on a laptop), and was not actively watching the pedestrians. Videos were recorded when experimenters had availability, between the hours of 8 am and 7 pm. The videos were recorded at different times of the day to reduce the likelihood that the same pedestrians were videotaped more than once. Since university classes occur on a Monday/Wednesday/Friday schedule, or a Tuesday/Thursday schedule, overlapping times were avoided across those days (i.e., if video was recorded at 8:00–10:00 am on Monday, that time interval was not recorded on Wednesday or Friday). Video recording continued until the semester ended.

## Participants

A total of 2,671 pedestrians were captured descending the staircases. Since the study focused on young adults, 271 pedestrians were excluded (161 midlife adults, 43 older adults, and 67 non-determined age). Thus, 2,400 young adults were included in the analyses (1,470 men (61%)); 1,013 pedestrians on the short staircase (484 men (48%)), and 1,387 pedestrians on the long staircase (986 men (71%)).

## Data coding

Trained coders analyzed the video recordings using PotPlayer (https://potplayer.daum.net/) with zoom features. The coders were graduate students in Health and Kinesiology; coders were taking a graduate level course that examined slips, trips, and falls across the adult lifespan.

Coding criteria were developed for behaviors identified as risky on stairs [18, 21, 24, 27–29, 31]; coding criteria (outlined below) were developed using a set of pilot videos in multiple locations. These locations included other campus locations, shopping malls, and public transportation stations (results from these pilot videos not included here). For each coded variable, there were two coders: primary and secondary. The primary coder coded the assigned behavior for all pedestrians. The secondary coder coded approximately 25% of the same pedestrians. The results of the two coders were used to quantify coding reliability [32].

**Pedestrian characteristics and footwear.** Coders quantified pedestrian characteristics including age group (young adults: 18–39 years, midlife adults: 40–60 years, or older adults: 61 + years), sex (man or woman), and footwear (flats, sandals, sliders, or high heels). Coding for age and sex was based on pedestrian appearance (e.g., facial features and anatomy) consistent with previous observational studies [18, 20–22, 33]. Footwear coding criteria were as follows: Flats had a flat footbed and the foot was fully enclosed by the shoe (e.g., running shoes). Sandals had a flat foot bed with straps at the toe and ankle. Sliders had a flat footbed with a toe strap but no ankle strap. Sliders are also known as flip flops, slides, or slip-ons. High heels included footwear with a raised heel.

**Pedestrian behaviors.** Coders quantified hand behaviors, gaze behaviors, interpersonal behaviors, and gait deviations, as outlined below. All behaviors were coded as 'yes' if they occurred at least once during the descent. In addition, coders identified handrail use and tread gaze behaviors at the top four or bottom four steps, called the transition steps. Behavior during the transition steps was coded because fall risk is higher during transition to/from a staircase [34] and visual attention to the transition steps is apparent due to gaze shifts or overt visual attention [35]. Since the short staircase only contained two steps, both were transition steps, and this distinction was not relevant.

**Hand behavior.** Hand behavior included four variables: handrail use, hands in pocket, carrying items, and electronic device use. Handrail use was identified if the pedestrian touched the handrail at least once while walking down the stairs (coded as yes/no). Handrail use on transition steps was identified separately if the pedestrian touched the handrail in the transition steps on the long staircase (yes/no). Hands in pocket was identified when at least one hand was in a pocket (yes/no). Carrying items was identified if carrying an item in at least one hand (yes/no). We coded when the person was using an electronic device such as a smartphone (yes/no).

**Gaze behavior.** Gaze behavior was coded by reviewing head angle [21], as head angle provides a valid surrogate assessment of actual gaze direction [36]. Coders observed each pedestrian's head angle to determine if their gaze would land on the stair tread. Note that the video software provided zoom features to facilitate viewing. Coders noted that pedestrians demonstrated multiple behaviors, including gaze on the treads, gaze on other pedestrians, gaze forward, etc., but coding was limited to pedestrians viewing the treads (except for the five pedestrians who lost balance, as described below). Tread gaze was coded if the gaze was located on the stair treads at least once during the descent (yes/no), and at least once on the transition steps (yes/no).

**Interpersonal behavior.** Two variables for interpersonal behavior included: Pedestrians were coded as walking beside another pedestrian (i.e., walking abreast [33]) (yes/no) and if they were having an in-person conversation (yes/no). In-person conversations were coded as 'yes' based on both (1) proximity of pedestrians walking abreast and (2) mouth movement of at least one of the dyad, indicating speaking,

**Gait deviations.** Gait deviations were defined as a change in the expected gait descent behavior. These deviations included skipping steps (yes/no) (pedestrian intentionally skipped alternate steps), stopping (yes/no), and loss of balance (yes/no). Anomalies (yes/no) were a

change in the expected behavior that resulted from another pedestrian, such as changing direction to avoid another pedestrian that had slowed down.

## Coding reliability

As noted above, a primary coder quantified stair-descent behavior for all captured pedestrians and a secondary coder coded approximately 25% of the same pedestrians. To quantify reliability between the primary and secondary coders, Cohen's kappa statistics were calculated and interpreted using Landis and Koch [32]: almost perfect ($\kappa$ range: 0.81–1.00), substantial (0.61–0.80), moderate (0.41–0.60), fair (0.21–0.40), slight (0.00–0.20) and poor agreement (<0.00). We excluded any variables that had less than moderate agreement. On the short staircase, almost perfect agreement was observed for all variables except for skipping steps ($\kappa$ = 0.50, moderate) and gaze behavior ($\kappa$ = 0.03, slight); gaze behavior on the short staircase was excluded due to slight agreement [Table 1]. On the long staircase, almost perfect and substantial agreement were observed in all variables except for skipping steps, which had moderate agreement.

## Co-occurrence of risky behaviors

To quantify the number of co-occurring risky behaviors, we computed the total number of risky behaviors for each pedestrian. Eight behaviors were considered risky behavior for this analysis: no handrail use, no tread gaze, footwear (sandals, sliders, or high heels), having an in-person or smartphone conversation, using an electronic device, having hands in pocket(s), using hands, and skipping steps. The behaviors on the transition steps were excluded here (handrail use and tread gaze), because they are a subset of the behaviors on the staircase. Thus, each pedestrian had a value ranging from 0 to 8 for co-occurrence of risky behaviors.

**Table 1. Inter-coder reliability expressed with Cohen's kappa and percent agreement.**

| Coded Variable | Short Staircase (2 steps) | | Long Staircase (17 steps) | |
|---|---|---|---|---|
| | Kappa | % Agreement | Kappa | % Agreement |
| Age | 0.83 | 98.0 | 0.89 | 98.1 |
| Sex | 0.99 | 99.7 | 1.00 | 99.7 |
| Footwear | 0.85 | 98.3 | 0.83 | 98.4 |
| Handrail use | **N/A | 99.7 | 0.94 | 90.0 |
| Handrail use transition steps | - | - | 0.91 | 86.2 |
| Hands in pocket | 0.82 | 90.2 | 0.95 | 98.1 |
| Carrying an item | 0.94 | 97.6 | 0.93 | 96.8 |
| Using an electronic device | 0.97 | 99.0 | 0.91 | 98.1 |
| Tread gaze | *0.03 | 53.2 | 0.76 | 95.7 |
| Tread gaze transition steps | - | - | 0.75 | 95.4 |
| Walking side-by-side | 1.00 | 100.0 | 1.00 | 96.5 |
| In-person conversation | 1.00 | 100.0 | 1.00 | 96.5 |
| Skipping steps | 0.50 | 99.3 | 0.45 | 97.8 |
| Stopping | **N/A | 99.7 | 0.75 | 99.2 |
| Loss of balance | 1.00 | 100.0 | 0.66 | 99.2 |
| Anomalies (change direction, hesitation, etc.) | 0.83 | 99.3 | **N/A | 99.7 |

* slight agreement; due to low values, tread gaze on the short staircase was excluded

** N/A: Cohen's kappa was not computed for these variables due to an unbalanced number of observed levels between coders (e.g., one coder observed two levels (yes and no) whereas the second coder only observed one level (no))

### Details regarding pedestrians who lost balance

The details regarding the circumstances and behavior of the pedestrians who lost balance were summarized in table format. These details included the eight coded risky behaviors (as in the preceding section) as well as: step number where balance was lost, recovery of balance, number of steps to recover balance, grabbed handrail, handrail within reaching distance, hand(s) available to grab handrail, gaze on staircase at time of balance loss, and gaze location at time of balance loss. To judge proximity of the handrail, anything ≥1.5 arm's length was considered unreachable. The handrail proximity and the number of steps to recover balance was coded by two coders for the pedestrians who lost balance; they had 100% agreement.

### Statistics

Descriptive statistics were reported as (1) number of pedestrians demonstrating the behavior and (2) percent of pedestrians demonstrating the behavior. Behaviors were also reported for men and women separately, also as number and percent. Statistical analysis was conducted using SAS 9.3 (Cary, NC, USA). For each coded behavior, odds ratio (OR) was calculated to determine differences between men and women during stair descent (men/women). For number of co-occurring risky behaviors, a two-sample t-test was used to analyze differences across men and women. With large sample sizes, parametric tests become robust to deviations in most of their key assumptions [37]. Significance level was p ≤ .05.

## Results

Results are presented first for all pedestrians, followed by results for men versus women.

### Footwear

On the short staircase, 92% of pedestrians (n = 931) wore flats, 5% (n = 52) wore sandals, 2% (n = 23) wore sliders, and 1% (n = 7) wore high heels [Table 2]. On the long staircase, 94% (n = 1,306) wore flats, 4% (n = 56) wore sandals, 1% (n = 15) wore sliders, and 1% (n = 10) wore high heels [Table 2]. On both staircases, a higher percentage of men wore flat shoes (short staircase: 95% and 89%, men and women, respectively; OR (95% CI): 2.22 (1.36–3.6l); long staircase: 97% and 87%; OR (95% CI): 5.21 (3.24–8.38)), and a lower percentage of men

**Table 2. Footwear of pedestrians quantified for all pedestrians combined (total) and separately for men and women.** Odds ratio (OR) and 95% confidence interval (CI) for men versus women.

| Staircase | Shoes | Observation as Percent (n) | | | OR (95% CI) | |
|---|---|---|---|---|---|---|
| | | Total (N = 1013) | Men (N = 484) | Women (N = 529) | Men vs Women | p |
| Short Staircase | Flat shoes | 92% (n = 931) | 95% (n = 459) | 89% (n = 472) | 2.217 (1.36–3.61) | **<0.01** |
| | Sandals | 5% (n = 52) | 2% (n = 9) | 8% (n = 43) | 0.214 (0.10–0.45) | **<0.01** |
| | Sliders | 2% (n = 23) | 3% (n = 16) | 1% (n = 7) | 2.549 (1.04–6.26) | 0.41 |
| | High heels | 1% (n = 7) | 0% (n = 0) | 1% (n = 7) | No comparison because no men wore heels | |
| Staircase | Shoes | Total (N = 1387) | Men (N = 986) | Women (N = 401) | Men vs Women | p |
| Long Staircase | Flat shoes | 94% (n = 1306) | 97% (n = 958) | 87% (n = 348) | 5.211 (3.24–8.38) | **<0.01** |
| | Sandals | 4% (n = 56) | 2% (n = 17) | 10% (n = 39) | 0.163 (0.09–0.29) | **<0.01** |
| | Sliders | 1% (n = 15) | 1% (n = 11) | 1% (n = 4) | 1.120 (0.35–3.54) | 0.85 |
| | High heels | 1% (n = 10) | 0% (n = 0) | 2% (n = 10) | No comparison because no men wore heels | |

Significant differences are bolded (p≤0.05).

wore sandals (short staircase: 2% and 8%; OR (95% CI): 0.21 (0.10–0.45); long staircase: 2% and 10%; OR (95% CI): 0.16 (0.09–0.29)) [Table 2].

## Hand behavior: Handrail use

None of the pedestrians used the handrail of the short staircase. Conversely, while descending the long staircase, 35% (n = 488) of pedestrians used the handrail at least once; men used the handrail more than women (37% and 31%, men and women, respectively; OR (95% CI): 1.33 (1.04–1.70); p = .03). In the transition steps of the long staircase, 33% (n = 463) of pedestrians used the handrail; men used the handrail more than women (35% and 29%, men and women, respectively; OR (95% CI): 1.36 (1.05–1.75); p = .02).

## Hand behavior: Hands in pocket and hand use

Men put their hands in their pocket more than women on both the short staircase (15% and 10%, men and women, respectively; OR (95% CI): 1.69 (1.16–2.48); p < .01) and the long staircase (21% and 9%, men and women, respectively; OR (95% CI): 2.71 (1.86–3.95); p < .01). Women used their hands to hold items (i.e., coffee cup or clothing) more than men on both the short staircase (63% and 78%, men and women, respectively; OR (95% CI): 0.48 (0.37–0.64); p < .01) and the long staircase (38% and 60%, men and women, respectively; OR (95% CI): 0.41 (0.32–0.52); p < .01). One couple on each staircase held hands during descent. On the short staircase, 16% of pedestrians (17% and 15%, men and women, respectively; OR (95% CI): 1.20 (0.86–1.67); p = .30) used electronic devices. On the long staircase, 12% of pedestrians used electronic devices while descending the stairs (12% and 12%, men and women, respectively; OR (95% CI): 0.96 (0.67–1.37); p = .81) [Table 3].

**Table 3. Observed behaviors in pedestrians, quantified for all pedestrians combined (total) and for men and women separately, in the short and long staircases.** All variables were coded yes/no, and percentage indicates numbers of participants with the behavior. Odds ratio (OR) and 95% confidence interval (CI) for men versus women.

| Staircase | Behavior | Observation as Percent (n) | | | OR (95% CI) | |
|---|---|---|---|---|---|---|
| | | Total (N = 1013) | Men (N = 484) | Women (N = 529) | Men vs Women | p |
| Short Staircase | In-person conversation | 16.4% (n = 166) | 13.6% (n = 66) | 18.9% (n = 100) | 0.677 (0.48–0.95) | **0.02** |
| | Walking side-by-side | 20.1% (n = 204) | 17.1% (n = 83) | 22.9% (n = 121) | 0.698 (0.51–0.95) | **0.02** |
| | Electronic device user | 16.1% (n = 163) | 17.4% (n = 84) | 14.9% (n = 79) | 1.196 (0.86–1.67) | 0.30 |
| | Handrail use | 0 | 0 | 0 | - | - |
| | Hands in pocket | 12.3% (n = 125) | 15.3% (n = 74) | 9.6% (n = 51) | 1.692 (1.16–2.48) | **<0.01** |
| | Hand use | 71.0% (n = 719) | 63.2% (n = 306) | 78.1% (n = 413) | 0.483 (0.37–0.64) | **<0.01** |
| Staircase | Behavior | Total (N = 1387) | Men (N = 986) | Women (N = 401) | Men vs Women | p |
| Long Staircase | In-person conversation | 14.5% (n = 201) | 13.3% (n = 131) | 17.5% (n = 70) | 0.724 (0.53–1.00) | **0.05** |
| | Walking side-by-side | 12.7% (n = 176) | 11.6% (n = 114) | 15.5% (n = 62) | 0.715 (0.51–1.00) | **0.05** |
| | Electronic device user | 11.9% (n = 165) | 11.8% (n = 116) | 12.2% (n = 49) | 0.958 (0.67–1.37) | 0.81 |
| | Handrail use | 35.2% (n = 488) | 37.0% (n = 365) | 30.7% (n = 123) | 1.328 (1.04–1.70) | **0.03** |
| | Handrail use during transition steps | 33.4% (n = 463) | 35.3% (n = 348) | 28.7% (n = 115) | 1.357 (1.05–1.75) | **0.02** |
| | Gaze on treads | 91.1% (n = 1263) | 90.2% (n = 889) | 93.3% (n = 374) | 0.662 (0.42–1.03) | 0.07 |
| | Gaze on treads during transition steps | 90.8% (n = 1260) | 89.9% (n = 886) | 93.3% (n = 374) | 0.633 (0.41–0.99) | **0.05** |
| | Hands in pocket | 17.6% (n = 244) | 21.1% (n = 208) | 9.0% (n = 36) | 2.711 (1.86–3.95) | **<0.01** |
| | Hand use | 44.2% (n = 613) | 37.8% (n = 373) | 59.9% (n = 240) | 0.408 (0.32–0.52) | **<0.01** |

Significant differences are bolded (p≤0.05).

## Gaze behavior

On the long staircase, 91% of pedestrians gazed in the direction of the stair tread at least once while descending; male gaze behavior was not different from female gaze behavior (90% and 93%, men and women, respectively; p = .07). During the transition steps, 91% of pedestrians gazed at the tread at the beginning and/or at the end of staircase; more women gazed at the transition steps than men (90% and 93%, men and women, respectively; OR (95% CI): 0.64 (0.41–1.03); p = .05) [Table 3].

## Interpersonal behavior

In-person conversations were significantly more frequent in women compared to men on both the short staircase (14% and 19%, men and women, respectively; OR (95% CI): 0.68 (0.48–0.95); p = .02) and the long staircase (13% and 18%; OR (95% CI): 0.72 (0.53–1.00); p = .05) [Table 3]. Since in-person conversations can only occur when walking with a colleague, and the number of women walking with others was significantly higher than men pedestrians in both staircases (p ≤ .05), the data was filtered to include only pedestrians walking with others (16% of participants, n = 380 for both staircases). In the filtered analysis, differences between men and women having a personal conversation were no longer significant for either staircase (p > .16).

## Gait deviations

Observations of gait deviations from each staircase were too low to calculate odd ratios for each staircase separately. Therefore, odd ratios were calculated on the data combined across the two staircases [Table 4]. Anomalies and stopping were not significantly different between men and women (anomalies: 0.4% and 0.4% for men and women, respectively; p = 0.94; stopping: 1.0% and 0.6% for men and women, respectively; p = .42). However, men were significantly more likely to skip steps than women (3.7% and 0.8%, men and women, respectively; OR (95% CI): 5.03 (2.28–11.10); p < .01).

**Table 4. Gait deviations for all pedestrians combined (total) and for men and women separately.** All variables were coded yes/no, and percentage indicates numbers of participants with the behavior. Since low numbers of incidents were observed at each individual staircase (less than 5), odds ratio (OR) and 95% confidence interval (CI) for men versus women were calculated for both staircases.

| Location | Behavior | Percent of observation (n) | | | OR (95% CI) | |
|---|---|---|---|---|---|---|
| | | **Total** | **Men** | **Women** | **Men vs Women** | *p* |
| Short staircase | Anomalies | 0.9% (n = 9/1013) | 1.0% (n = 5/484) | 0.8% (n = 4/529) | - | - |
| | Skipping steps | 2.0% (n = 20/1013) | 3.5% (n = 17/484) | 0.6% (n = 3/529) | - | - |
| | Stopping | 0.7% (n = 7/1013) | 1.0% (n = 5/484) | 0.4% (n = 2/529) | - | - |
| | Loss of balance | 0.1% (n = 1/1013) | 0.0% (n = 0/484) | 0.2% (n = 1/529) | - | - |
| Long staircase | Anomalies | 0.1% (n = 1/1387) | 0.1% (n = 1/986) | 0.0% (n = 0/401) | - | - |
| | Skipping steps | 3.0% (n = 41/1387) | 3.8% (n = 37/986) | 1.0% (n = 4/401) | - | - |
| | Stopping | 0.9% (n = 13/1387) | 0.9% (n = 9/986) | 1.0% (n = 4/401) | - | - |
| | Loss of balance | 0.3% (n = 4/1387) | 0.4% (n = 4/986) | 0.0% (n = 0/401) | - | - |
| Total | Anomalies | 0.4% (n = 10/2400) | 0.4% (n = 6/1470) | 0.4% (n = 4/930) | 0.949 (0.27–3.38) | 0.94 |
| | Skipping steps | 2.5% (n = 61/2400) | 3.7% (n = 54/1470) | 0.8% (n = 7/930) | 5.028 (2.28–11.10) | **<0.01** |
| | Stopping | 0.8% (n = 20/2400) | 1.0% (n = 14/1470) | 0.6% (n = 6/930) | 1.481 (0.57–3.87) | 0.42 |
| | Loss of balance | 0.2% (n = 5/2400) | 0.3% (n = 4/1470) | 0.1% (n = 1/930) | - | - |

Significant differences are bolded (p≤0.05).

Loss of balance was infrequent for both men and women (0.3% and 0.1% for men and women, respectively) and therefore, odd ratios were not calculated. Five pedestrians (four men, one woman) lost their balance, one woman on the short staircase and four men on the long staircase; all five recovered their balance and no falls were observed. None of the five pedestrians were using the handrail or looking at the stairs at the time of balance loss. Four of the five were completing another task at the time: one was texting with both hands, two were looking at the phone (perhaps texting with one hand), and one was putting an item in a bag. After losing balance, only one pedestrian grabbed the handrail (and then continued texting). All five looked at the stairs after losing balance.

### Co-occurring risky behaviors

The number of co-occurring risky behaviors in the pedestrians ranged from 0 to 6 (maximum possible was 8) (Fig 2A). The majority of pedestrians (69.2%) had two or more risky behaviors, and few pedestrians had zero risky behaviors (9.6%). The majority of men (62.2%) had two or more risky behaviors, and 13% had zero risky behaviors. The vast majority of women (80.3%) had two or more risky behaviors, and 4.3% had zero risky behaviors (Fig 2B). Men had fewer co-occurring risky behaviors than women (1.9 vs 2.3, men vs women, respectively; t(2162.2) = 8.33, p<0.001).

### Pedestrians who lost balance

Five pedestrians lost their balance, and all five recovered their balance (Table 5). Four of the five pedestrians were men, all the men lost balance on the long staircase, the woman lost balance on the short staircase. Four of the pedestrians lost balance at the top step, and one lost balance midway on the long staircase. Mean number of co-occurring risky behaviors in the pedestrians was 2.6, and mean number of steps to recover balance was 1.4 steps. Only one pedestrian grabbed the handrail during balance recovery. This pedestrian was able to move quickly enough to stop texting and grab the handrail. At the time of balance loss, all five pedestrians were looking at something other than the stairs.

### Discussion

Previous research has demonstrated that average injury rate of stair-related falls has a trimodal distribution with a peak for young adults in their 20's, with women in their 20's sustaining

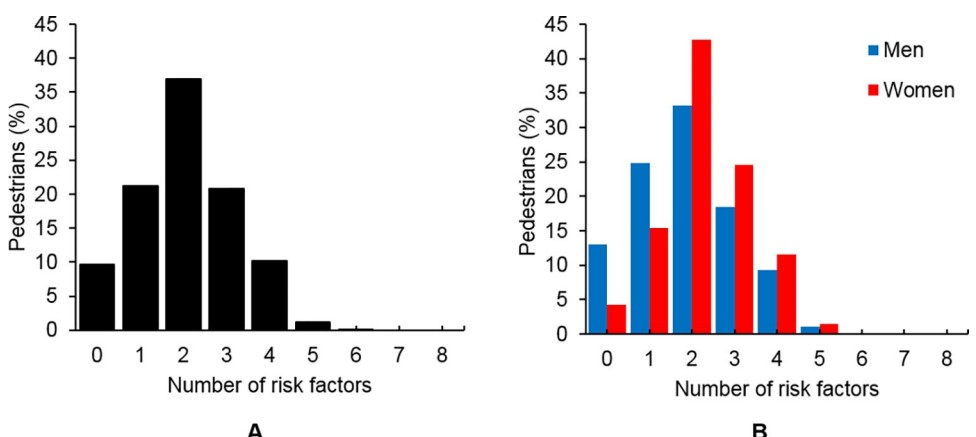

**Fig 2.** Histogram of the number of risk behaviors for all pedestrians combined (A) and for men and women separately (B).

**Table 5. Circumstances and behaviors in the five pedestrians who lost their balance during stair descent.**

| Characteristic/ Behavior | Pedestrian | | | | |
|---|---|---|---|---|---|
| | A | B | C | D | E |
| Sex | Male | Male | Male | Male | Female |
| Staircase | Long | Long | Long | Long | Short |
| In-person conversation | No | No | No | No | No |
| Skipping steps | No | No | No | No | No |
| Electronic device user | No | Yes | No | Yes | Yes |
| Handrail use | No | No | No | No → Yes, after losing balance | No |
| Gaze on treads (anytime during descent) | Yes | Yes | Yes | Yes | Yes |
| Hands in pocket | No | No | No | No | No |
| Hand use | Yes | Yes | No | Yes | Yes |
| Footwear | Sandals | Flats | Flats | Flats | Flats |
| Total number risky behaviors | 3 | 3 | 1 | 3 | 3 |
| Stair number where balance was lost | 1 (top) | 7 (middle) | 1 (top) | 1 (top) | 1 (top) |
| Balance recovered? | Yes | Yes | Yes | Yes | Yes |
| Number of steps to recover balance | 2 | 1 | 2 | 1 | 1 |
| Grabbed handrail? | No | No | No | Yes | No |
| Handrail within reaching distance? | No | Yes | Yes | Yes | No |
| Hand(s) available to grab handrail? | No, both hands unavailable | No, hand on handrail—side unavailable | Yes | No, but stopped texting and grabbed handrail | No, hand on handrail—side unavailable |
| Gaze on staircase when balance lost? | No | No | No | No | No |
| Gaze location at time of balance loss | Gaze on bag (putting item in bag) | Gaze to left (lab area*) | Gaze to left (lab area*) | Gaze on smartphone | Gaze on smartphone |

* An engineering lab with glass windows is located to the left of the staircase. Pedestrians B & C were gazing at the lab when they lost their balance. Bolded, red text indicates risky behaviors during stair descent.

about 80% more injuries than men of the same age [16]. The purpose of this study was to quantify young adult behaviors during staircase descent, with the goal of identifying behaviors that increase fall risk on stairs, and to determine if young women are more likely to demonstrate these risky behaviors. Young adults engaged in behaviors that may increase fall risk, including not using the handrail, carrying items, using electronic devices, holding a conversation, and wearing footwear that increases fall risk. Women demonstrated several behaviors that increase their fall risk relative to men: less likely to use the handrail, more likely to be holding something in their hands, more likely to be engaged in conversation, more likely to wear sandals and heels, and they demonstrated a higher number of co-occurring risky behaviors. However, women also engaged in behaviors that may be protective of fall risk: less likely to skip steps and more likely to look at the stair tread during transition steps. Overall, the observed sex-related differences are strengthened by the inclusion of both short and long staircases since similar sex-related effects were observed on both staircases.

If a person loses their balance, a handrail can aid in balance recovery when standing on the ground [38] and when standing on stairs [39]. However, in the present study, no one used the handrail on the short staircase, consistent with observations on a three-step staircase in laboratory research (participants included older adults and young adults with a simulated vision disability) [40, 41]. Not using the handrail on short staircases is especially relevant because there is a high occurrence of falls on stairs with less than five steps [9, 30]. On the long staircase, there is an increase in injury severity if a fall occurs [9, 30], but the majority of young adults (65%) did not use the handrail. This finding is consistent with previous observational research (range: 57–76%) [18–21], but less than the 94% observed in one study [22]. The higher rate in the latter study is most likely due to data collection during the height of the COVID-19 pandemic [22], while the current dataset and the other cited studies were collected before the pandemic. Cultural norms and physical proximity were altered during the pandemic, which likely reduced handrail use in [22]. Overall, the majority of young adults did not use handrails proactively (i.e., holding the handrail while descending, before a loss of balance occurs). However, laboratory research demonstrates that a reactive response occurs when balance is lost on the stairs: the hand rapidly moves toward an external support and sufficient force is generated to prevent a fall on stairs [39]. It is important to note that in the Maki, Perry [39] study, participants *expected* their balance to be disturbed, but the timing and magnitude of the perturbation were unpredictable. When a fall on stairs in the lab was *unexpected* due to deception, not all young adult participants tried to grasp the handrail, and grasp errors were observed in participants who tried [42]. In the current study, we also observed that four of five young adults who lost their balance did not grasp the handrail. Thus, while young adults have the ability to reactively grasp and use the handrail to recover their balance when they are expecting a loss of balance, they are less successful when the loss to balance is unexpected, such as would occur outside the lab setting. Therefore, not using the handrail in a proactive manner is risky–and women were less likely to use the handrail than men. Lower proactive handrail use in women may explain, at least in part, the high stair-related injury rate in young adult women [16], especially since women have slower and more variable reaction times [43] and are slower to grasp a handrail in a reactive task when walking in a lab setting [44].

Potential reasons for not using the handrail include: (1) using the hands for another purpose, (2) having hands in pockets, (3) walking with another person (when the other person is beside the handrail), (4) being too far from the handrail, and (5) concerns about cleanliness of the handrails. Hand use prevents proactive use of the handrail as observed, for example, in four of the five pedestrians who lost balance (pedestrians A, B, D, E; Table 5). Three of these four were using an electronic device, consistent with previous stair observation research that handrail use was reduced when using a mobile device [22]. Further, holding an item also

inhibits the ability to reactively grasp the handrail [38]. The tendency to keep hold of an item despite a loss of balance has even been observed in infants, demonstrating the pervasiveness of maintaining hold on an item [45]. Four of the five pedestrians who lost balance had items in their hands, and three of them did not grab the handrail. The fourth pedestrian was texting with both hands, he stopped texting and grabbed the handrail with one hand while maintaining hold of the phone in his other hand (pedestrian D). In both staircases, the majority of women used their hands to hold items (78.1% and 59.9% of women on the short and long staircases, respectively). Conversely, while men were more likely than women to have their hands in their pockets, this behavior was observed in the minority of participants (15.3% and 21.1% of men on the short and long staircases, respectively). When walking with someone, one pedestrian would be between the other pedestrian and the handrail, preventing grasping the handrail. Women were more likely to be walking with someone, and thus would be more likely to be prevented from grasping the handrail; however, walking with someone was observed in the minority of participants (22.9% and 15.5% of women on the short and long staircases, respectively). Finally, we consider that people may avoid using handrails because they are frequently touched by others; high-touch surfaces may be contaminated by microorganisms, leading to transmission of infections [46]. Since women appear to be more concerned about personal hygiene than men (women wash their hands more frequently, are more likely to use soap, and wash their hands for a longer duration [47, 48]), the higher percentage of women not using the handrail may also be associated with gender-related differences in personal hygiene.

Falls on stairs can occur when a person is distracted [9]; the young adults here were distracted by using electronic devices (13.7% of all pedestrians) and by having conversations (15.3% of all pedestrians). Stair descent requires visual monitoring of the environment and control of trunk and limb movements, with appropriate foot trajectories that clear each stair edge and land safely on each stair tread [49–51]. Using an electronic device reduces attention, alters gaze behavior, reduces the visual field, and impairs gait on stairs [28, 29, 52, 53]. Similarly, when gaze is diverted by a cognitive task on stairs, gait is altered [22, 35, 54]. These impairments result in the increasing pedestrian injuries due to mobile phone use [26]. Previous research demonstrated that half of the falls on stairs in young adults were related to texting [5]. Thus, high injury rates of stair-related falls in young adults [16] may be related to the use of electronic devices. However, since men and women were not different in their use of electronic devices when descending stairs, this behavior does not explain the higher number of stair-related injuries in women unless women are more impaired/distracted by the use of electronic devices than men. The effect of sex during locomotor multitasks in young adults has not been extensively studied; two papers indicate that young men and women were not impaired differently during the following locomotor multitasks: cognitive task during timed-up-and-go task [55], and cognitive task when using a treadmill desk [56]. Thus, there is no evidence that young adult women are more impaired than men by locomotor multitasks, but future studies should continue to examine sex-related differences among young adults in a wider range of locomotor multitasks.

Young adult women were more likely to engage in conversations during stair descent, which may be related to the higher injury rate of stair-related falls for women. During conversations, the pedestrian is either listening or speaking; both tasks are continuous and cognitively demanding. Listening involves cognitive, affective, and behavioral processes; the listener attends to, understands, and interprets verbal and non-verbal cues [57]. Speech requires memory, selection of appropriate words and grammar, and the coordination of breathing and speech patterns [58, 59]. During challenging multitasks, young healthy adults flexibly allocate resources between the tasks [59, 60], and tasks with higher values are prioritized [61, 62].

Young adults may prioritize the conversation task, as people with impaired speech are perceived as incompetent [63, 64]. This possibility is supported by lab-based research that observed gait impairment during stair negotiation when young adults have a phone conversation [27]. Similarly, in fall survey research in young adults, the most common secondary task at the time of a fall was talking to a friend [4, 5], and women were more likely than men to report talking to a friend at the time of a fall [5]. A recent observational study on stair negotiation also found that women were more likely to be distracted (distraction included one or more of the following: Mobile device (looking at, talking on, or holding), using earbuds or headphones, talking with a peer [22].

While women were more likely to engage in conversation than men, it may be related to the observation that women were more likely to walk with a colleague. Being more likely to walk with a colleague is consistent with the previous research that women often interact at a closer intimate distance with colleagues than men [65, 66], and female dyads walk abreast more often than male dyads [33]. Since an in-person conversation requires a colleague, we completed a follow-up analysis where people who were not walking with a colleague were excluded. After this exclusion, there was no difference in talking with a colleague between men and women. Thus, while women were more likely to have an in-person conversation, this was driven by the greater likelihood that the women walked with a colleague.

The vast majority of pedestrians (91%) looked at the steps of the long staircase at least once, likely to gather visual information regarding stair features (i.e., position of stair edge, dimensions of the risers and treads) and drive motor behavior [23]. In the long staircase, women were more likely to gaze at the steps, which may be protective for falls. Women were also more likely to look at the tread in a recent observational study on stairs in a public setting [21]. For loss of balance, we note that none of the five pedestrians who lost balance were looking at the stairs when their balance was lost, and all of them looked at the stairs during the recovery. The pedestrians were looking at the following items rather than the stairs: smartphone (two pedestrians), the surrounding environment (two pedestrians), and looking at clothing item being put in bag (one pedestrian). Four of the five pedestrians were at the top/transition step when they lost balance, consistent with previous research [34]. On the long staircase, lighting was brightest at the transition step so loss of balance in these cases does not appear to be related to lighting levels. The apparent role of gaze diversion in the loss of balance is consistent with two observational studies that demonstrated that not looking down at the step appeared to be associated with falls [24] and more stair incidents were observed with infrequent tread gaze [21]. Therefore, gaze behavior and its association with falls on stairs should be examined further.

Seven percent of young adults wore shoes that are known to increase fall risk and injuries in midlife and older adults: sandals, sliders, and high heels. In older adults, footwear is a risk factor for falls [67, 68]. In adults aged 45 years and over, sandals and sliders increased the risk of fall-related foot fractures and medium- to high-heel shoes increased the risk of fractures at multiple sites [69]. In young adults, lab-based research indicates impaired balance when young adults wear sliders or heels. Sliders resulted in longer heel slip distance and velocity during slips relative to industry standard slip resistant shoes [70]. High heels compromised balance and induced changes to the neuromechanics of gait that were widespread and mostly disadvantageous [71, 72]. Regarding sex differences, young women may sustain more injuries on stairs due to their footwear: Only women wore high heels, and women were more than four times as likely to wear sandals. In addition, many students wear comfortable shoes due to the size of the campus; in settings such as a professional workplace or social gatherings, women may be even more likely to wear heels.

Skipping steps was observed more frequently in men; skipping steps is a deliberate, intentional act and implies confidence during stair descent. However, stumbling and/or falling may

result if they misjudge the step position. Skipping steps was mostly observed at the bottom of the staircase, consistent with previous reports [9, 73, 74]. Therefore, although the risk of falling may be higher with this behavior, the impact forces will be reduced when falling from the bottom versus top of the stairs [9, 30]. We note that the fall risk associated with skipping steps may be compounded by the lower number of men that gazed at the tread in the transition steps; future studies should identify if those skipping steps are more or less likely to gaze at the transition steps.

The majority of young adult pedestrians demonstrated co-occurring risky behaviors during stair descent, with young women demonstrating a higher mean number of risky behaviors than men (1.9 vs 2.3, men vs women, respectively; Fig 2). Therefore, although women demonstrated both risky behavior and protective behavior, the overall result was a higher number of risky behaviors for women than men. As noted in earlier text, the higher number of co-occurring behaviors in women appear to result from behaviors associated with gender. These gender-specific behaviors include avoiding high-touch surfaces (i.e., handrails), footwear choices, and being more likely to walk with a colleague and thus more likely to be engaged in in-person conversation [33, 47, 48]. Stair safety strategies should target the risky behaviors most commonly observed: not using the handrail and hands being unavailable to grasp the handrail if needed. While people should be encouraged to use the handrail proactively and not to carry items in their hands, the prevalence of electronic devices and to-go coffees will likely impair compliance. Therefore, we propose task-specific training where participants practice reactive handrail grasping following a perturbation while safely harnessed (i.e., the stair-perturbation protocol from Maki, Perry [39]). This training is similar to the promising approach to fall prevention where slips and/or trips are applied during walking to effectively train neuromuscular balance responses [75, 76]. Slip and trip training is effective after a single training session, so it is possible that short training sessions will promote successful handrail grasping. Handrail grasping training should include the participants holding an item, since grasping the handrail is inhibited when holding an item [38]. This training may improve movement time and success in grasping the handrail. Further, environmental modifications would be based on identifying handrail characteristics that improve balance recovery such as handrail height [77] and handrail size/shape [78, 79]. Similarly, increasing handrail visibility may be effective, since hand muscle activity increased when participants simply viewed a safety handle [80]. Increasing the visibility of the stairs or making the stairs appear larger through illusions improves stair gait behavior [81]; these modifications may also help mitigate the effects of distracted attention due to electronic devices and in-person conversations.

There are several limitations that warrant consideration. First, the variables age and sex were coded based on participant appearance. While coders demonstrated good agreement (Table 1), this may result from both coders being incorrect since both coders used the same appearance to make judgements. However, judgement based on appearance is the only method available to identify age and sex in observational/field research and has been used previously [18, 20–22, 33]. Second, pedestrians walking behind others were not coded due to visual obstruction. People who walk close behind others may demonstrate risky behaviors that were not captured in this study. Third, staircase observations were limited to a university campus setting. Typical university behaviors (e.g., walking to classes at frequent intervals, higher density of people on stairs at certain times of day) may not be similar in other settings/populations, and the results may not translate to young adults in settings outside of campus. Fourth, most stair-related injuries occur at home [16], but it was not possible to observe people in their homes. Fifth, no women in the study were visibly pregnant, and pregnant women are at a high risk of falls [82]. Sixth, the risk factors were summed where each risk factor contributed equally, which assumes that each factor contributed equally to fall risk. However, it is likely

that some factors connote greater risk than others. but we did not have a basis for determining weighting of each factor, so we opted for the simple summation. Finally, the illumination on the stairs was affected by the weather and time of day. The variability in factors such as lighting is a common issue for observational studies, and will increase variability. We did not collect lighting level at the same time as the pedestrians were videotaped, but this should be included in future observational studies to identify the effect of lighting.

In conclusion, young adults demonstrated multiple behaviors that are likely to increase fall risk during stair descent. These behaviors include: not using the handrail, carrying items, using electronic devices, holding a conversation, and wearing footwear that increases fall risk. Risky behaviors observed more frequently in women include: less likely to use the handrail, more likely to be holding something in their hands, more likely to be engaged in conversation, more likely to wear sandals and heels, and demonstrating a higher number of co-occurring risky behaviors. Risky behaviors observed more frequently in men include: more likely to skip steps and less likely to look at the stair tread during transition steps. These behaviors may be related to the high injury rate of stair-related falls in young adults, and the higher injury rate observed in young women versus men [16]. The observations on the pedestrians who lost their balance indicate that the top step appears to be particularly risky, as well as gaze diverted away from the stairs.

## Acknowledgments

The authors thank Marissa Munoz-Ruiz and Lucas J. Rooney for participating in helpful discussions when the project was conceived.

## Author Contributions

**Conceptualization:** HyeYoung Cho, Amanda J. Arnold, Chuyi Cui, Zihan Yang, Tim Becker, Ashwini Kulkarni, Anvesh Naik, Shirley Rietdyk.

**Data curation:** HyeYoung Cho, Ashwini Kulkarni.

**Formal analysis:** HyeYoung Cho, Amanda J. Arnold.

**Investigation:** HyeYoung Cho, Amanda J. Arnold, Chuyi Cui, Zihan Yang, Tim Becker, Ashwini Kulkarni, Anvesh Naik, Shirley Rietdyk.

**Methodology:** HyeYoung Cho, Amanda J. Arnold, Chuyi Cui, Zihan Yang, Tim Becker, Ashwini Kulkarni, Anvesh Naik, Shirley Rietdyk.

**Project administration:** HyeYoung Cho, Shirley Rietdyk.

**Supervision:** HyeYoung Cho, Shirley Rietdyk.

**Writing – original draft:** HyeYoung Cho, Amanda J. Arnold, Chuyi Cui, Zihan Yang, Tim Becker, Ashwini Kulkarni, Anvesh Naik, Shirley Rietdyk.

**Writing – review & editing:** HyeYoung Cho, Amanda J. Arnold, Chuyi Cui, Zihan Yang, Tim Becker, Ashwini Kulkarni, Anvesh Naik, Shirley Rietdyk.

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
