## [Decision Letter · Decision Letter 0]

22 Mar 2023

PONE-D-23-02948Risky behaviors in young adults during stair descent: males versus femalesPLOS ONE

Dear Dr. Rietdyk,

Thank you for submitting your manuscript to PLOS ONE. After careful consideration, we feel that it has merit but does not fully meet PLOS ONE’s publication criteria as it currently stands. Therefore, we invite you to submit a revised version of the manuscript that addresses the points raised during the review process.

We look forward to receiving your revised manuscript.

Kind regards,

In-Ju Kim, Ph.D.

Academic Editor

PLOS ONE

Journal Requirements:

Additional Editor Comments:

Reviewer No. 1:

General comments

This study investigated the risky behaviours for a large group (N=2400) of young adults when descending two types of staircase (a short 2 step staircase and a longer 17 step staircase) in a real-world/naturalistic setting. This was a significant undertaking to report the stair user behaviour and builds nicely on previous empirical studies based in the laboratory environment. The authors should be commended for this.

The article describes risky behaviours for young adults in general but pays particular attention to the differences between males and females. The presentation of results are informative and easy to digest, and the discussion of each risk behaviour are in depth and consider very relevant literature. One aspect I felt the article is currently lacking is a description or reporting of any interacting factors. I would assume that some trials are a combination of risky behaviours (e.g. all or some of the following - no handrail use, using an electronic device, gaze behaviour characteristics and/or footwear). Can the authors comment on this, and where appropriate, include further details in the discussion please.

Images are used in the methods to illustrate the staircase design. Are the authors able to share additional images (exemplar scenarios) or video footage as supplementary material that enables the reader to visualise the stair user in some of the scenarios listed (e.g. hand behaviour, head angle, interpersonal behaviours, gait deviations)? This would really enrich the article.

The limitations are signposted clearly and allow the reader to make a fair judgement of the findings together with existing literature. Overall the manuscript is well written, clear, and detailed. I do not have any serious concerns, but I do have a few comments in addition to the broad comments above, mostly related to clarity.

General abstract comments:

The abstract should make clear the difference between staircases size, and clearly highlight where the risky behaviours were present on the short, long or both staircases.

The first sentence of results focuses on both male and female young adults combined, but I would suggest rewriting in the context of differences between females and males as this appears to be the primary purpose of the article.

General Introduction comments:

Very clear and appropriate rationale developed throughout.

Line 88 – do you know the illumination for each staircase? For the 17-step staircase do you know how consistent the lighting was?

Line 89 – you have provided information for the riser height, tread depth and stair width. For long staircases there is typically some variation in dimensions between steps. Can you comment on whether there is some variation on your staircase or if each step were measured individually and all had the same dimensions? If variation exists, please include this in your description. This may be relevant for any falls reported in the results section.

Related to the information included already, other built environment features are missing from the description that are apparent in Figure 1 (e.g. handrail height and type, presence and type of stair nosing, edge highlighter presence, location, and colour/contrast). These details should be included in the text (or figure caption at least) since they are not easy to discern from the images and all influence stair safety.

The long staircase is described as having 17 steps but Figure 1B shows that there are more, presumably you coded the bottom half of the staircase, but why was this chosen instead of the top half? Please justify your decision in the text.

Line 103 – how did you determine that data capture should occur for six weeks? Was this based on logistical reasons or was testing halted as soon as you reached a specific number of stair users?

Line 104 – did lighting vary depending on the time of day? If so, please describe how and/or include stair tread illumination. Lighting can influence risky behaviour on stairs.

Line 112 – it would be very interesting to see how the midlife and older adult data that you captured differs compared to the young adults. This seems beyond the scope of the current paper but would be interesting as a follow up article, especially in the context of understanding how ageing across the life span affects stair behaviour.

Line 133 – “called” rather than “call”.

Line 134 – “is” rather than “if”.

Line 149 – how did you subjectively review head angle? Were guidelines/instructions provided to each coder to ensure consistency? Or were images digitised to quantify head angle?

Line 168 – please provide the threshold for levels of agreement, to place the table results in to context.

Line 189 – I would suggest removing “more likely” as this wording could seem like the pedestrians chose what footwear to wear based on the staircase, which is unlikely/not the case. You could state the percentage difference or that more males wore flat shoes than females etc.

Line 192 – Table 2. Female column percentage = 99%. Is there a rounding error?

Line 229 – the text formatting on this line appears to be different to the main body of text.

Line 248 – Is it possible to determine how many recovery steps – albeit subjective – each pedestrian took until they “recovered” their balance? Although the number of participants (N=5) is low, this data is fascinating and can really help with future risk prediction models.

Line 296 – in the discussion relating to the 5 pedestrians who did not grasp the handrail to recover balance, can you comment on their proximity to the handrail. Stair width was ~1.8m thus it may not have been feasible to reach/grasp the handrail, and this should be considered in your nice discussion around handrail use.

Line 299 – new paragraph started but still with the same context as previous paragraph same I would suggest reformatting here.

Line 412 – remove “be” from this line.

Line 416, yes handrail height, but also handrail design/shape. See the fantastic work by Vicki Komisar, Phillipa Gosine and Alison Novak in this area to add further context.

Reviewer No. 2:

I would like to appreciate the authors for conducting a study on observing risk behaviors during stair descent in male and female young adults in a college setting. There are several methodological concerns that must be addressed before considering the manuscript for further review. Please see below my comments and suggestions to improve the content and clarity of the manuscript.

Title: Use “males” and “females” as adjectives and “men” and “women” as nouns throughout the manuscript.

Abstract

Only line 28 reports results! Explain how the study was done before summarizing results.

Line 28 Report number of men and women

p≤0.05 should be p<0.05, the threshold for statistical significance

Lines 72-74 The objectives are not very clear – Were there two objectives? Was the primary objective to identify young adults' (risky?) behaviors on stairs? Was the secondary objective to compare the risky behaviors on stairs between sexes?

Please add the relevant hypotheses after aims.

Lines 74 to 80 Should be under the methods section.

Line 83 were the pedestrians videotaped? If so, there must be an informed consent signed by the participants?

Line 90 Add camera manufacturer details

Line 101 Please add “Behavioral variables and coding criteria” as an appendix/supplementary material

Line 118 Please add more information on qualification and relevant experiences of the coders.

Line 121 “quantified” should read as “guessed”

Line 123-124 Need further details on how these characteristics were coded to be replicated by future studies.

Why were all coders’ codes not included in the reliability study? Please re-analyze reliability by randomly selecting a few participants and comparing the codes from all coders for these participants and then update Table 1 accordingly.

Lines 180-182 Add how descriptive statistics were summarized. Why did the authors not run the Chi-square tests first and then look at odd ratios for only those variables from the comparisons that turned out statistically significant in the Chi-square tests?

Under the results section, first descriptively summarize the frequency of risky and no risky behaviors during stair descent observed in male and female young adults as well those of all participants (both sexes combined). Then specifically discuss the differences between both sexes for each behavior.

Odds ratio interpretation is inadequate! The authors generally stated males more than females for each behavior rather than exactly pointing out how much more based on the odds ratios. This results section must be revised.

After discussing the limitations of the study in the discussion section, please add future recommendations.

Line 441-443 the following sentences are not based on the current study -

“high injury rate of stair related falls in young adults, and the higher injury rate observed in young adult females versus males”. This should be excluded.

Reviewers' comments:

Reviewer's Responses to Questions

**Comments to the Author**

1. Is the manuscript technically sound, and do the data support the conclusions?

Reviewer #1: Yes

Reviewer #2: Partly

2. Has the statistical analysis been performed appropriately and rigorously? 

Reviewer #1: Yes

Reviewer #2: No

3. Have the authors made all data underlying the findings in their manuscript fully available?

Reviewer #1: Yes

Reviewer #2: Yes

4. Is the manuscript presented in an intelligible fashion and written in standard English?

Reviewer #1: Yes

Reviewer #2: Yes

5. Review Comments to the Author

Reviewer #1: General comments

This study investigated the risky behaviours for a large group (N=2400) of young adults when descending two types of staircase (a short 2 step staircase and a longer 17 step staircase) in a real-world/naturalistic setting. This was a significant undertaking to report the stair user behaviour and builds nicely on previous empirical studies based in the laboratory environment. The authors should be commended for this.

The article describes risky behaviours for young adults in general but pays particular attention to the differences between males and females. The presentation of results are informative and easy to digest, and the discussion of each risk behaviour are in depth and consider very relevant literature. One aspect I felt the article is currently lacking is a description or reporting of any interacting factors. I would assume that some trials are a combination of risky behaviours (e.g. all or some of the following - no handrail use, using an electronic device, gaze behaviour characteristics and/or footwear). Can the authors comment on this, and where appropriate, include further details in the discussion please.

Images are used in the methods to illustrate the staircase design. Are the authors able to share additional images (exemplar scenarios) or video footage as supplementary material that enables the reader to visualise the stair user in some of the scenarios listed (e.g. hand behaviour, head angle, interpersonal behaviours, gait deviations)? This would really enrich the article.

The limitations are signposted clearly and allow the reader to make a fair judgement of the findings together with existing literature. Overall the manuscript is well written, clear, and detailed. I do not have any serious concerns, but I do have a few comments in addition to the broad comments above, mostly related to clarity.

General abstract comments:

The abstract should make clear the difference between staircases size, and clearly highlight where the risky behaviours were present on the short, long or both staircases.

The first sentence of results focuses on both male and female young adults combined, but I would suggest rewriting in the context of differences between females and males as this appears to be the primary purpose of the article.

General Introduction comments:

Very clear and appropriate rationale developed throughout.

Line 88 – do you know the illumination for each staircase? For the 17-step staircase do you know how consistent the lighting was?

Line 89 – you have provided information for the riser height, tread depth and stair width. For long staircases there is typically some variation in dimensions between steps. Can you comment on whether there is some variation on your staircase or if each step were measured individually and all had the same dimensions? If variation exists, please include this in your description. This may be relevant for any falls reported in the results section.

Related to the information included already, other built environment features are missing from the description that are apparent in Figure 1 (e.g. handrail height and type, presence and type of stair nosing, edge highlighter presence, location, and colour/contrast). These details should be included in the text (or figure caption at least) since they are not easy to discern from the images and all influence stair safety.

The long staircase is described as having 17 steps but Figure 1B shows that there are more, presumably you coded the bottom half of the staircase, but why was this chosen instead of the top half? Please justify your decision in the text.

Line 103 – how did you determine that data capture should occur for six weeks? Was this based on logistical reasons or was testing halted as soon as you reached a specific number of stair users?

Line 104 – did lighting vary depending on the time of day? If so, please describe how and/or include stair tread illumination. Lighting can influence risky behaviour on stairs.

Line 112 – it would be very interesting to see how the midlife and older adult data that you captured differs compared to the young adults. This seems beyond the scope of the current paper but would be interesting as a follow up article, especially in the context of understanding how ageing across the life span affects stair behaviour.

Line 133 – “called” rather than “call”.

Line 134 – “is” rather than “if”.

Line 149 – how did you subjectively review head angle? Were guidelines/instructions provided to each coder to ensure consistency? Or were images digitised to quantify head angle?

Line 168 – please provide the threshold for levels of agreement, to place the table results in to context.

Line 189 – I would suggest removing “more likely” as this wording could seem like the pedestrians chose what footwear to wear based on the staircase, which is unlikely/not the case. You could state the percentage difference or that more males wore flat shoes than females etc.

Line 192 – Table 2. Female column percentage = 99%. Is there a rounding error?

Line 229 – the text formatting on this line appears to be different to the main body of text.

Line 248 – Is it possible to determine how many recovery steps – albeit subjective – each pedestrian took until they “recovered” their balance? Although the number of participants (N=5) is low, this data is fascinating and can really help with future risk prediction models.

Line 296 – in the discussion relating to the 5 pedestrians who did not grasp the handrail to recover balance, can you comment on their proximity to the handrail. Stair width was ~1.8m thus it may not have been feasible to reach/grasp the handrail, and this should be considered in your nice discussion around handrail use.

Line 299 – new paragraph started but still with the same context as previous paragraph same I would suggest reformatting here.

Line 412 – remove “be” from this line.

Line 416, yes handrail height, but also handrail design/shape. See the fantastic work by Vicki Komisar, Phillipa Gosine and Alison Novak in this area to add further context.

Reviewer #2: I would like to appreciate the authors for conducting a study on observing risk behaviors during stair descent in male and female young adults in a college setting. There are several methodological concerns that must be addressed before considering the manuscript for further review. Please see below my comments and suggestions to improve the content and clarity of the manuscript.

Title: Use “males” and “females” as adjectives and “men” and “women” as nouns throughout the manuscript.

Abstract

Only line 28 reports results! Explain how the study was done before summarizing results.

Line 28 Report number of men and women

p≤0.05 should be p<0.05, the threshold for statistical significance

Lines 72-74 The objectives are not very clear – Were there two objectives? Was the primary objective to identify young adults' (risky?) behaviors on stairs? Was the secondary objective to compare the risky behaviors on stairs between sexes?

Please add the relevant hypotheses after aims.

Lines 74 to 80 Should be under the methods section.

Line 83 were the pedestrians videotaped? If so, there must be an informed consent signed by the participants?

Line 90 Add camera manufacturer details

Line 101 Please add “Behavioral variables and coding criteria” as an appendix/supplementary material

Line 118 Please add more information on qualification and relevant experiences of the coders.

Line 121 “quantified” should read as “guessed”

Line 123-124 Need further details on how these characteristics were coded to be replicated by future studies.

Why were all coders’ codes not included in the reliability study? Please re-analyze reliability by randomly selecting a few participants and comparing the codes from all coders for these participants and then update Table 1 accordingly.

Lines 180-182 Add how descriptive statistics were summarized. Why did the authors not run the Chi-square tests first and then look at odd ratios for only those variables from the comparisons that turned out statistically significant in the Chi-square tests?

Under the results section, first descriptively summarize the frequency of risky and no risky behaviors during stair descent observed in male and female young adults as well those of all participants (both sexes combined). Then specifically discuss the differences between both sexes for each behavior.

Odds ratio interpretation is inadequate! The authors generally stated males more than females for each behavior rather than exactly pointing out how much more based on the odds ratios. This results section must be revised.

After discussing the limitations of the study in the discussion section, please add future recommendations.

Line 441-443 the following sentences are not based on the current study -

“high injury rate of stair related falls in young adults, and the higher injury rate observed in young adult females versus males”. This should be excluded.

Thank you for the opportunity to review the manuscript.

6. PLOS authors have the option to publish the peer review history of their article (what does this mean?). If published, this will include your full peer review and any attached files.

Reviewer #1: **Yes: **Richard Foster

Reviewer #2: No

---

## [Author Response · Author response to Decision Letter 0]

25 May 2023

We thank the reviewers for their time and expertise in completing this review. We have addressed all the comments as outlined below. Major changes resulting from addressing the comments include: a new measure with an associated new figure (co-occurrence of risky behaviors observed in each pedestrian), and a complete description in table format of the five pedestrians who lost their balance. We believe that addressing the comments has improved the clarity and contribution of this manuscript. 

Please note that line numbers refer to the version with track changes.

Reviewer No. 1:

General comments

This study investigated the risky behaviours for a large group (N=2400) of young adults when descending two types of staircase (a short 2 step staircase and a longer 17 step staircase) in a real-world/naturalistic setting. This was a significant undertaking to report the stair user behaviour and builds nicely on previous empirical studies based in the laboratory environment. The authors should be commended for this.

The article describes risky behaviours for young adults in general but pays particular attention to the differences between males and females. The presentation of results are informative and easy to digest, and the discussion of each risk behaviour are in depth and consider very relevant literature. One aspect I felt the article is currently lacking is a description or reporting of any interacting factors. I would assume that some trials are a combination of risky behaviours (e.g. all or some of the following - no handrail use, using an electronic device, gaze behaviour characteristics and/or footwear). Can the authors comment on this, and where appropriate, include further details in the discussion please.

RESPONSE: We thank the reviewer for their review and for their positive comments in support of our work. Regarding combination of risky behaviors, thank you for highlighting this factor! We agree that this information is relevant, and have quantified the number of co-occurring risky behaviors for each pedestrian; see Figure 2. We did not look at patterns of co-occurrence as that is beyond the scope of this paper, but the data would be suitable to big data approaches, such as unsupervised learning. We also included a more thorough description of the risky behaviors exhibited by the five pedestrians that lost their balance in a new table, Table 5. Please see text added for the new co-occurrence analyses (see lines 243-250, 351-358, 533-541 for methods, results, and discussion), and the five pedestrians that lost balance (see line 251-259, 363-376, 432-435, 501-507 for methods, results, and discussion) 

Images are used in the methods to illustrate the staircase design. Are the authors able to share additional images (exemplar scenarios) or video footage as supplementary material that enables the reader to visualise the stair user in some of the scenarios listed (e.g. hand behaviour, head angle, interpersonal behaviours, gait deviations)? This would really enrich the article.

RESPONSE: We carefully considered making video clips available. When reviewing the video clips for this purpose, it was evident that if someone knew the videotaped pedestrian, they would recognize them even though the face was blurred. We were concerned because losing balance is generally perceived negatively in young adults. If we blurred other parts of the body, the behavior would not be evident. Since the pedestrians were students/staff at Purdue University (or at least the vast majority of videotaped pedestrians were students/staff), they are more likely to be identified than people videotaped at another public location, such as a shopping mall. Several people on the team thought that these concerns would not be a problem given the ubiquity of cameras in our communities, while others felt that they themselves would be upset if they were on the video that was shared publicly, and also was shared by faculty/grad students of the university. Thus, since we did not have explicit permission from the pedestrians to post videos or photos, we decided to err on the side of protecting the privacy of the pedestrians and did not post any videos or photos. 

The limitations are signposted clearly and allow the reader to make a fair judgement of the findings together with existing literature. Overall the manuscript is well written, clear, and detailed. I do not have any serious concerns, but I do have a few comments in addition to the broad comments above, mostly related to clarity.

General abstract comments:

The abstract should make clear the difference between staircases size, and clearly highlight where the risky behaviours were present on the short, long or both staircases.

 RESPONSE: Changes made as requested. See lines 27-44

The first sentence of results focuses on both male and female young adults combined, but I would suggest rewriting in the context of differences between females and males as this appears to be the primary purpose of the article.

RESPONSE: We were also interested in quantifying behavior for both sexes since stair-related injuries demonstrates a trimodal distribution, with a peak for people in their 20’s (the other peaks are ≤3 years and ≥85 years). To clarify this, we have edited the purpose statement as follows in the abstract and introduction: “The purposes of this observational study were (1) to quantify young adult behaviors during stair descent and (2) to identify any differences between males and females.” See lines 28-30, 82-83

General Introduction comments:

Very clear and appropriate rationale developed throughout.

Line 88 – do you know the illumination for each staircase? For the 17-step staircase do you know how consistent the lighting was? See our response below.

Line 89 – you have provided information for the riser height, tread depth and stair width. For long staircases there is typically some variation in dimensions between steps. Can you comment on whether there is some variation on your staircase or if each step were measured individually and all had the same dimensions? If variation exists, please include this in your description. This may be relevant for any falls reported in the results section. See our response below.

Related to the information included already, other built environment features are missing from the description that are apparent in Figure 1 (e.g. handrail height and type, presence and type of stair nosing, edge highlighter presence, location, and colour/contrast). These details should be included in the text (or figure caption at least) since they are not easy to discern from the images and all influence stair safety.

RESPONSE: Changes made as requested. We have added details to the methods to more fully describe the staircases, illumination, and handrails. We have added new photos to figure 1 to provide a more complete view of the staircases. See lines 98-131

The long staircase is described as having 17 steps but Figure 1B shows that there are more, presumably you coded the bottom half of the staircase, but why was this chosen instead of the top half? Please justify your decision in the text.

RESPONSE: The bottom half was selected for video visibility reasons. Text added as requested, see lines 111-113

Line 103 – how did you determine that data capture should occur for six weeks? Was this based on logistical reasons or was testing halted as soon as you reached a specific number of stair users?

RESPONSE: In the previous semester, we had collected multiple videos on various staircases to develop our coding criteria, and to determine staircases that had a high amount of pedestrian traffic, good visibility, good lighting, and a location to unobtrusively videotape. Once our criteria were developed, we finalized our staircase selection, and then we recorded video until the end of the academic semester. See lines 138-150, 161-169 

Line 104 – did lighting vary depending on the time of day? If so, please describe how and/or include stair tread illumination. Lighting can influence risky behaviour on stairs.

RESPONSE: We have added text to demonstrate the lighting change for the short staircase (line 108) and the long staircase (lines 127-131) 

Line 112 – it would be very interesting to see how the midlife and older adult data that you captured differs compared to the young adults. This seems beyond the scope of the current paper but would be interesting as a follow up article, especially in the context of understanding how ageing across the life span affects stair behaviour.

RESPONSE: Yes, we agree. Our pilot video data collection included more pedestrians at midlife and older because they were collected at a shopping mall and other locations off-campus, but we decided to focus on young adults due to the trimodal distribution of stair-related injuries, with a peak in the 20’s. We highly recommend the development of automated coding for future observational studies! No changes made. 

Line 133 – “called” rather than “call”.

Line 134 – “is” rather than “if”.

RESPONSE: Changes made as requested.

Line 149 – how did you subjectively review head angle? Were guidelines/instructions provided to each coder to ensure consistency? Or were images digitised to quantify head angle?

RESPONSE: Images were not digitized. The coders were instructed to determine if the gaze would land on the stair treads based on the head angle. We acknowledge that this is a subjective assessment. To facilitate consistency, the following approach was used: (1) two coders were designated as gaze behavior coders (a primary coder coded 100% of trials, and a secondary coder coded 25% of trials), (2) when coders focus on one or two behaviors, precision of behavioral coding improves, and (3) we tested reliability between coders as outlined in the methods and results. This is the same approach other coders have used and recent research has demonstrated that head angle provides a valid assessment of gaze direction (see line 200-201). We have added text to more clearly describe our coding instructions for head angle, see lines 201-206

Line 168 – please provide the threshold for levels of agreement, to place the table results in to context.

RESPONSE: Text has been added as requested, see lines 227-234 

Line 189 – I would suggest removing “more likely” as this wording could seem like the pedestrians chose what footwear to wear based on the staircase, which is unlikely/not the case. You could state the percentage difference or that more males wore flat shoes than females etc.

RESPONSE: Text has been edited for clarity on line 277-280

Line 192 – Table 2. Female column percentage = 99%. Is there a rounding error?

RESPONSE: Yes, we have a rounding error because, in the table, we reported percentages as whole numbers, as suggested by Cole (2015): “…if the range is 10% or more use whole numbers, if less than 1% use two decimal places, and otherwise one.”

Cole TJ. Too many digits: the presentation of numerical data. Arch Dis Child. 2015 Jul;100(7):608-9. 

Line 229 – the text formatting on this line appears to be different to the main body of text.

RESPONSE: We agree that it appears different in the pdf, so we reformatted in Word, but nothing changed. It appears to be an illusion? 

Line 248 – Is it possible to determine how many recovery steps – albeit subjective – each pedestrian took until they “recovered” their balance? Although the number of participants (N=5) is low, this data is fascinating and can really help with future risk prediction models.

RESPONSE: We have coded and added this data, see lines 363-371 and Table 5. 

Line 296 – in the discussion relating to the 5 pedestrians who did not grasp the handrail to recover balance, can you comment on their proximity to the handrail. Stair width was ~1.8m thus it may not have been feasible to reach/grasp the handrail, and this should be considered in your nice discussion around handrail use.

RESPONSE: We have coded and added this data, see lines252-259 and Table 5. We also added this as a reason for not using the handrail (line 425).

Line 299 – new paragraph started but still with the same context as previous paragraph same I would suggest reformatting here.

RESPONSE: Changes made as suggested.

Line 412 – remove “be” from this line.

RESPONSE: Change made as suggested.

Line 416, yes handrail height, but also handrail design/shape. See the fantastic work by Vicki Komisar, Phillipa Gosine and Alison Novak in this area to add further context.

 RESPONSE: Thank you for highlighting this fascinating work! We had seen their perturbation on stairs, but not the effect of handrail design on withstanding perturbations. We have added text to the discussion to include their work on handrail design, see lines 556

Reviewer No. 2:

I would like to appreciate the authors for conducting a study on observing risk behaviors during stair descent in male and female young adults in a college setting. There are several methodological concerns that must be addressed before considering the manuscript for further review. Please see below my comments and suggestions to improve the content and clarity of the manuscript.

RESPONSE: Thank you for your review and your positive comments. We have addressed the concerns below.

Title: Use “males” and “females” as adjectives and “men” and “women” as nouns throughout the manuscript.

RESPONSE: Changes made as requested.

Abstract

Only line 28 reports results! Explain how the study was done before summarizing results.

Line 28 Report number of men and women

RESPONSE: Changes made as requested to abstract 

p≤0.05 should be p<0.05, the threshold for statistical significance

RESPONSE: Both p<0.05 and p≤0.05 are commonly used thresholds; we have always used the latter. Note as well, that in the abstract when we use p≤0.05 at the end of the sentence that includes of a list of variables, we are using the traditional format to indicate that all the variables listed in the sentence are statistically significant at p≤0.05. 

Lines 72-74 The objectives are not very clear – Were there two objectives? Was the primary objective to identify young adults' (risky?) behaviors on stairs? Was the secondary objective to compare the risky behaviors on stairs between sexes?

RESPONSE: We apologize for the confusion. To increase clarity, we have used numbers in the sentence to indicate there are two objectives as you describe above. See lines 28-30 in the abstract and lines 82-83 in the introduction.

Please add the relevant hypotheses after aims.

RESPONSE: Because this was an observational study, we did not have any hypotheses. 

Lines 74 to 80 Should be under the methods section.

RESPONSE: The text here includes our rationale for observing and quantifying behavior during stair descent, on indoor staircases (long and short), and university campus staircases. In our opinion, this rationale belongs in the introduction rather than the methods. Therefore, we did not change the location. 

Line 83 were the pedestrians videotaped? If so, there must be an informed consent signed by the participants?

RESPONSE: Yes, the pedestrians were videotaped. The University’s IRB determined that our research was exempt from IRB review and consent forms were not required, please see text 93-96. For more information, research can be considered exempt under several categories. Our research fell under category 2: “Research that only includes interactions involving educational tests (cognitive, diagnostic, aptitude, achievement), survey procedures, interview procedures or observation of public behavior (including visual or auditory recording).” Please note that prior to submitting the IRB exemption request, we researched what is appropriate and legal regarding video recordings in order to maintain the rights of the observed pedestrians. From the website for the American Civil Liberties Union, we learned the following: “Taking photographs and video of things that are plainly visible in public spaces is a constitutional right.” (Text cited from: aclu.org/issues/free-speech/photographers-rights/filming-and-photographing-police#:~:text=Taking%20photographs%20and%20video%20of,officials%20carrying%20out%20their%20duties). 

Line 90 Add camera manufacturer details

RESPONSE: Changes made as requested, please see lines 138

Line 101 Please add “Behavioral variables and coding criteria” as an appendix/supplementary material

RESPONSE: We apologize for the confusion. The coding criteria was included later (after line 101), but the location of the text that you note (line 101 from the previous version) was misplaced. We have modified the methods so that it is apparent that the coding criteria is provided in the methods. Please see lines 160-222. 

Line 118 Please add more information on qualification and relevant experiences of the coders.

RESPONSE: Changes made as requested, please see lines 161-169(Dating coding section). 

Line 121 “quantified” should read as “guessed”

RESPONSE: Wording was changed from “quantified” to “coded”

Line 123-124 Need further details on how these characteristics were coded to be replicated by future studies.

RESPONSE: We have added more details as requested throughout the “Data Coding” section. We also note that we believe our organization of the previous version was problematic with some details in the wrong section. The organization has also been improved. See lines 160-221

Why were all coders’ codes not included in the reliability study? Please re-analyze reliability by randomly selecting a few participants and comparing the codes from all coders for these participants and then update Table 1 accordingly.

RESPONSE: We apologize for the confusion; we have edited the text to increase clarity. While there were six coders in total, each behavior was coded by only two coders. Therefore, all coders’ codes were included in the reliability assessment; however, the reliability was calculated by comparing the codes from the two coders that contributed to the coding of that specific behavior. For example, only two coders examined tread gaze behavior on the two staircases. Given the difficulty and time-consuming nature of behavioral coding, it is common practice to utilize multiple coders but only designate two coders (a primary and secondary coder) for each behavior to assess reliability. By allowing coders to solely focus on one or two behaviors, there is an increase in the precision of behavioral coding. For examples of research using multiple coders to examine specific behaviors, with two coders coding the same behavior please see Rachwani et al. (2020) and Arnold & Claxton (in press, Developmental Psychology). 

See edits on lines 161-169 to increase clarity.

Rachwani, J., Kaplan. B. E., Tamis-LeMonda, C. S., Adolph, K. E. (2021). Children’s' use of everyday artifacts: Learning the hidden affordance of zipping. Developmental Psychobiology, 63(4), 739-799. https://doi.org/10.1002/dev.22049.

Arnold, A. J. & Claxton, L. J. (in press). Effect of carrying objects on walking characteristics and language abilities in 13- and 24-month-olds. Developmental Psychology. DOI: 10.1037/dev0001535

Lines 180-182 Add how descriptive statistics were summarized. Why did the authors not run the Chi-square tests first and then look at odd ratios for only those variables from the comparisons that turned out statistically significant in the Chi-square tests?

RESPONSE: The method for the descriptive statistics has been added, see lines 261-268 (statistics section). We are not familiar with the approach of conducting first a Chi-square followed by an odds ratio (OR) on the significant results. The odds ratio test provides statistical significance, so conducting the Chi-square first appears redundant. Perhaps the two-step approach (Chi-square then OR) the reviewer describes is used when there are more than two groups/levels to compare? The approach we used is consistent with published research that compared falls in older women and men (e.g. Yang et al., 2018) and our own work with falls in young women versus young men (Cho et al., 2021). This latter citation includes co-author Dr. Bruce Craig, who is director of the Statistics Consulting Service at Purdue. Thus, we believe we have correctly completed the odds ratio analyses. 

Yang, et al. "Sex differences in the circumstances leading to falls: Evidence from real-life falls captured on video in long-term care." Journal of the American Medical Directors Association 19.2 (2018): 130-135.

Cho, et al. "Falls in young adults: The effect of sex, physical activity, and prescription medications." PLoS one 16.4 (2021): e0250360.

Under the results section, first descriptively summarize the frequency of risky and no risky behaviors during stair descent observed in male and female young adults as well those of all participants (both sexes combined). Then specifically discuss the differences between both sexes for each behavior.

Odds ratio interpretation is inadequate! The authors generally stated males more than females for each behavior rather than exactly pointing out how much more based on the odds ratios. This results section must be revised.

RESPONSE: We apologize for the confusion which we believe stemmed from the lack of clarity in our purpose. We have updated our purpose statement based on the reviewers’ comments to reflect the following: 1) quantify risk behaviors in young adults on stairs, and 2) compare behavior of men versus women. Thus, we report the combined results first (i.e., men and women combined). The text first presents a descriptive summary of the combined results. Then we report any significant differences between men and women, including the OR and the CI. The percent change, OR, and CI were missing from the footwear section, and we have corrected this omission (lines 277-280). Note that we did not report the OR in the text when it was not significant. We trust that these changes satisfy this issue. 

After discussing the limitations of the study in the discussion section, please add future recommendations.

 RESPONSE: Throughout the discussion, we have described future research with the relevant text. For example, in the section on gaze behavior, at the end of the paragraph (line 510) the following text is included: “Therefore, gaze behavior and its association with falls on stairs should be examined further.” Future recommendations are also provided in lines 467-468, 510, 531-532, and the paragraph on perturbation training (546-561). This is an acceptable approach that some researchers adopt (embedding the future recommendations with the text associated with the interpreted variables), and we prefer to keep these future recommendations with the associated text.

Line 441-443 the following sentences are not based on the current study -

“high injury rate of stair related falls in young adults, and the higher injury rate observed in young adult females versus males”. This should be excluded.

RESPONSE: Thank you for noting that this statement is not based on the current study. We have added the reference associated with this statement. We did not exclude the statement as injuries on stairs in young adults was a primary rationale for completing the study.

---

## [Decision Letter · Decision Letter 1]

28 Jun 2023

Risky behavior during stair descent for young adults: differences in men versus women

PONE-D-23-02948R1

Dear Dr. Rietdyk,

We’re pleased to inform you that your manuscript has been judged scientifically suitable for publication and will be formally accepted for publication once it meets all outstanding technical requirements.

Kind regards,

In-Ju Kim, Ph.D.

Academic Editor

PLOS ONE

Additional Editor Comments (optional):

Reviewers' comments:

Reviewer's Responses to Questions

**Comments to the Author**

1. If the authors have adequately addressed your comments raised in a previous round of review and you feel that this manuscript is now acceptable for publication, you may indicate that here to bypass the “Comments to the Author” section, enter your conflict of interest statement in the “Confidential to Editor” section, and submit your "Accept" recommendation.

Reviewer #1: All comments have been addressed

Reviewer #2: All comments have been addressed

2. Is the manuscript technically sound, and do the data support the conclusions?

Reviewer #1: Yes

Reviewer #2: Yes

3. Has the statistical analysis been performed appropriately and rigorously? 

Reviewer #1: Yes

Reviewer #2: Yes

4. Have the authors made all data underlying the findings in their manuscript fully available?

Reviewer #1: Yes

Reviewer #2: Yes

5. Is the manuscript presented in an intelligible fashion and written in standard English?

Reviewer #1: Yes

Reviewer #2: Yes

6. Review Comments to the Author

Reviewer #1: The authors have done a tremendous job addressing both sets of previous reviewer comments. The response letter and changes to manuscript are thorough throughout, well done.

I am particularly pleased to see the addition of co-occurring risk behaviours and an indepth description of the circumstances surrounding the 5 participants who lost their balance on the stairs. Table 5 is a very nice addition that adds value to the results of the manuscript.

This paper is written to a very high standard and will be a noteworthy addition to the exisiting stair safety literature.

Reviewer #2: I would like to congratulate the authors for addressing the reviewers comments.

Regarding hypothesis:

Please note: “Observational or interventional studies should have a hypothesis for choosing research design and sample size. The results of observational and interventional studies further lead to the generation of new hypotheses, testing of which forms the basis of future studies.” Cf. doi: 10.3346/jkms.2021.36.e338

Many thanks for the opportunity to review the manuscript.

7. PLOS authors have the option to publish the peer review history of their article (what does this mean?). If published, this will include your full peer review and any attached files.

Reviewer #1: No

Reviewer #2: No

---

## [Editor Report · Acceptance letter]

3 Jul 2023

PONE-D-23-02948R1 

Risky behavior during stair descent for young adults: differences in men versus women 

Dear Dr. Rietdyk:

I'm pleased to inform you that your manuscript has been deemed suitable for publication in PLOS ONE. Congratulations! Your manuscript is now with our production department. 

Kind regards, 

on behalf of

Dr In-Ju Kim 

Academic Editor

PLOS ONE